# Intraocular pressure elevation after subtenon triamcinolone acetonide injection; Multicentre retrospective cohort study in Japan

Yuki Maeda[1,2], Hiroto Ishikawa🄳[1,2]*, Hiroki Nishikawa[3], Miho Shimizu[1,4], Takamasa Kinoshita🄳[1,4], Rie Ogihara[1,5], Shigehiko Kitano[1,5], Chihiro Yamanaka[1,6], Yoshinori Mitamura[1,6], Masahiko Sugimoto[1,7], Mineo Kondo[1,7], Yoshihiro Takamura[1,8], Nahoko Ogata[1,9], Tomohiro Ikeda[1,2], Fumi Gomi[1,2]

1 J-CREST (Japan Clinical REtina STudy group), Kagoshima, Japan, 2 Department of Ophthalmology, Hyogo College of Medicine, Nishinomiya, Japan, 3 Center for Clinical Research and Education, Hyogo College of Medicine, Nishinomiya, Japan, 4 Department of Ophthalmology, Sapporo City General Hospital, Sapporo, Japan, 5 Department of Ophthalmology, Diabetes Center, Tokyo Women's Medical University School of Medicine, Shinjuku-ku, Japan, 6 Department of Ophthalmology, Tokushima University, Tokushima, Japan, 7 Department of Ophthalmology, Mie University, Tsu, Japan, 8 Department of Ophthalmology, Fukui University, Yoshida, Japan, 9 Department of Ophthalmology, Nara Medical University School of Medicine, Kashihara, Japan

* ohmyeye@gmail.com

**Data Availability Statement:** All relevant data are within the paper and its Supporting Information files.

## Abstract

### Purpose

To evaluate real-world evidence for intraocular pressure (IOP) elevation after subtenon triamcinolone acetonide injection (STTA) in 1252 Japanese patients (1406 eyes) in the Japan Clinical REtina STudy group (J-CREST).

### Methods

This was a multicentre retrospective study of the medical records of 1252 patients (676 men (758 eyes); mean age: 63.8 ± 12.9 years) who received STTA in participating centres between April 2013 and July 2017.

### Results

IOP elevation was observed in 206 eyes (14.7%) and IOP increase ≥ 6 mmHg was found in 328 eyes (23.3%). In total, 106 eyes (7.5%) needed medication and two eyes (0.14%) needed surgical procedures. Younger age, higher baseline IOP, and steroid dose were risk factors associated with IOP elevation. Risk factors associated with IOP increase ≥ 6 mmHg were younger age, lower baseline IOP, steroid dose, and higher incidences of diabetic macular oedema (DME) and uveitis. In contrast, with steroid dose fixed at 20 mg, a lower incidence of DME was a risk factor for increased IOP, suggesting that STTA had dose-dependent effects on IOP increase, especially in patients with DME.

**Funding:** The authors received no specific funding for this work.

**Competing interests:** The authors have declared that no competing interests exist.

## Conclusion

Our real-world evidence from a large sample of Japanese patients who received STTA showed that the incidence of IOP elevation after STTA was 14.7%, and was associated with younger age, higher baseline IOP, and steroid dose. Thus, IOP should be monitored, especially in patients with younger age, higher baseline IOP, and higher incidences of DME and uveitis.

## Introduction

Subtenon triamcinolone acetonide injection (STTA) was initially reported as treatment for optic neuritis [1], and has been used to treat diabetic macular oedema (DME) [2–10], cystoid macular oedema (CME) due to retinal vein occlusion (RVO) [11–13], uveitis [14–17], scleritis [18, 19], neuroretinitis [20], and CME following intraocular surgery [21]. Previous studies have shown that intravitreal triamcinolone acetonide injection (IVTA) is effective in patients with DME [22], CRVO [23], and BRVO [24]; however, endophthalmitis after IVTA was reported in rare cases [25]. In addition, two studies revealed that intraocular pressure (IOP) elevation was observed in 33%–50% and 59%–83% of patients who received IVTA [26, 27]. The risk of severe side effects due to STTA is presumably lower than the risk due to IVTA [4]; moreover, the STTA technique is easier than the IVTA technique. Therefore, Japanese ophthalmologists commonly choose STTA as initial treatment for both DME and RVO. The incidence of side effects is lower after STTA than after IVTA; however, side effects after STTA include IOP elevation, cataract formation [28], and central serous chorioretinopathy [29].

Steroid-induced IOP elevation was first described in the 1950s, with IOP surveillance following administration of systemic steroids [30, 31]. Potential complications of STTA, such as IOP elevation, were described in the 1970s [32]; this problem was discussed very actively in the 1990s [16, 33]. The presence of uveitis, younger patient age, and higher baseline IOP have been identified as risk factors for IOP elevation following STTA [34–37]; however, studies thus far have shown heterogeneous results and there is no consensus. The purpose of this study was to perform an analysis of real-world evidence in Japanese patients who exhibited IOP elevation following STTA, to determine the incidence of IOP elevation in relation to patient characteristics.

## Materials and methods

### Study design and eligibility

This was a multicentre retrospective study involving the following institutions (J-CREST): Hyogo College of Medicine, Sapporo City General Hospital, Tokyo Women's Medical University School of Medicine, Tokushima University, Tsukuba University, Mie University, Fukui University, and Nara Medical University School of Medicine. Between April 2013 and July 2017, a total of 1252 patients who received STTA were enrolled at the participating institutions. The current study was performed in accordance with the Declaration of Helsinki and with approval from the ethics committee of Hyogo College of Medicine (2421) and the ethics committees of the other participating hospitals.

### Patients

At each hospital in the J-CREST group, patients who received STTA were analysed using data from medical records. The observation period after STTA was > 6 months for all analysed

subjects. Exclusion criteria for this study were: 1) existing glaucoma diagnosis and associated medication; 2) baseline IOP > 21 mmHg; 3) any intraocular surgery except glaucoma surgery within 6 months after STTA injection. A total of 1406 eyes were analysed in 1252 Japanese patients who received STTA.

## Subtenon triamcinolone acetonide injection (STTA)

STTA was performed as follows: 20 mg of triamcinolone acetonide (TA) was injected into the subtenon space after topical anaesthesia. Based on each patient's eye conditions, STTA was performed at the discretion of each participating ophthalmologist; more than 90% of patients received STTA 1 time (20 mg total), while the remaining patients received it up to 5 times (20 mg per injection)

## Study protocol

Data were extracted from medical records in the various hospitals and sent to the data centre in the Department of Ophthalmology, Hyogo College of Medicine. The analysed data were as follows: age, sex, ratio of bilateral or unilateral STTA, baseline and highest IOP during follow-up periods, causative diseases, incidence of IOP elevation, incidence of IOP increase $\geq 6$ mmHg, total dose of TA, and treatments for IOP elevation. Causative diseases were categorized as DME, CME due to RVO, uveitis, and others. For analysis, patients were grouped on the basis of bilateral or unilateral STTA, with or without IOP elevation, and with or without IOP increase $\geq 6$ mmHg.

## IOP measurements

IOP measurements were performed using a Goldman tonometer or non-contact type tonometer in each hospital. IOP measurement in each patient was performed using the same device throughout the study. Baseline IOP was defined as the most recent IOP before STTA; the highest IOP was defined as the highest IOP after STTA during the follow-up period. When IOP elevation was observed, treatments (e.g., eye drops, oral medication, and surgery) were performed at the discretion of each participating ophthalmologist. IOP elevation was defined as IOP > 21 mmHg. IOP increase $\geq 6$ mmHg was also assessed as an additional measurement of IOP [34, 38–40].

## Study endpoints

The primary endpoint was the cumulative incidence of patients with IOP elevation (IOP > 21 mmHg) after STTA. The secondary endpoint was the incidence of patients with IOP increase $\geq 6$ mmHg. In addition, this study compared patients with bilateral STTA and unilateral STTA, patients with and without IOP elevation, and patients with and without IOP increase $\geq 6$ mmHg. Risk factors for IOP elevation and IOP increase $\geq 6$ mmHg were investigated based on the patients' characteristics.

## Statistical analyses

For continuous variables, the mean, standard deviation, median, and range were calculated. For discrete variables, the number of values in each category and the percentages in each category were calculated. To assess group differences, the Wilcoxon signed-rank test were used for continuous variables and Fisher's exact test were used for categorical variables. Analyses were performed with JMP® Pro (version 14.0.0, SAS Institute Inc., Cary, NC, USA). For all

analyses, p-values were reported, as were two-sided 95% confidence intervals for point estimates. Differences with $p < 0.05$ were considered statistically significant.

## Results

### Patients' demographics

The baseline characteristics in this study (1252 patients; 1406 eyes) and analyses of differences between bilateral and unilateral STTA are shown in Table 1. Briefly, patient age (mean ± standard deviation) was 63.8 ± 12.9 years; the age of patients with bilateral STTA was significantly younger than that of patients with unilateral STTA (p<0.0001, Wilcoxon signed-rank test). In total, 686 patients were men (54.8%). In all patients, baseline IOP was 14.2 ± 3.1 mmHg and the highest IOP during the follow-up period was 18.1 ± 5.0 mmHg. Causative diseases were DME (632 eyes, 45.0%), RVO (457 eyes, 32.5%), uveitis (223 eyes, 15.9%), and others (94 eyes, 6.7%), including optic neuritis, thyroid-associated ophthalmopathy, and Irvine-Gass Syndrome. The highest IOP values in patients with DME, RVO, uveitis, and others were 18.2 ± 4.5 mmHg, 18.1 ± 5.4 mmHg, 18.1 ± 5.7 mmHg, and 17.3 ± 4.8 mmHg, respectively. The incidences of DME and uveitis were significantly higher in patients with bilateral STTA than in patients with unilateral STTA (p<0.0001, Fisher's exact test). Sex, baseline and highest IOP, incidence of IOP elevation, and incidence of IOP increase ≥ 6 mmHg were not associated with bilateral STTA.

### Primary endpoint: IOP elevation (IOP > 21 mmHg)

Analyses of IOP elevation are shown in Table 2. Briefly, the incidence of IOP elevation (IOP > 21 mmHg) was 14.7% (206 eyes). The average interval between the last STTA and observation of IOP elevation was 77.4 ± 65.0 days (range: 1–405 days). The age in patients with IOP elevation was significantly younger than that in patients without IOP elevation (p<0.0001, Wilcoxon signed-rank test). Baseline and highest IOP, as well as total steroid dose, were significantly higher in patients with IOP elevation than in patients without IOP elevation (each p<0.0001, Wilcoxon signed-rank test). For treatments in patients with IOP elevation, eye drops were administered in 106 eyes (51.5% of eyes with IOP elevation, 7.5% of all eyes)

**Table 1. Patient characteristics.**

|  | Total | Patients with bilateral STTA | Patients with unilateral STTA | P-value |
|---|---|---|---|---|
| Number of patients | 1252 patients; 1406 eyes | 154 patients; 308 eyes | 1098 patients; 1098 eyes |  |
| Age | 63.8 ± 12.9 years | 61.2 ± 12.6 years | 64.6 ± 12.9 years | *<0.0001* |
| Sex (male) | 686 (54.8%) | 82 (53.3%) | 594 (54.1%) | 0.84 |
| Baseline IOP | 14.2 ± 3.1 mmHg | 14.3 ± 2.9 mmHg | 14.2 ± 3.1 mmHg | 0.63 |
| Highest IOP | 18.1 ± 5.0 mmHg | 17.9 ± 4.3 mmHg | 18.1 ± 5.2 mmHg | 0.71 |
| Incidence of IOP elevation | 206/1406 eyes (14.7%) | 37/308 eyes (12.0%) | 169/1098 eyes (15.4%) | 0.14 |
| Incidence of IOP increase ≥ 6 mmHg | 328/1406 eyes (23.3%) | 70/308 eyes (22.7%) | 258/1098 eyes (23.5%) | 0.78 |
| Causative diseases |  |  |  |  |
| DME | 632 eyes (45.0%) | 190 eyes (61.7%) | 442 eyes (40.3%) | *<0.0001* |
| RVO | 457 eyes (32.5%) | 29 eyes (9.4%) | 428 eyes (39.0%) |  |
| Uveitis | 223 eyes (15.9%) | 76 eyes (24.7%) | 147 eyes (13.4%) |  |
| Others | 94 eyes (6.7%) | 13 eyes (4.2%) | 81 eyes (7.4%) |  |

DME, diabetic macular oedema; IOP, intraocular pressure; RVO, retinal vein occlusion; STTA, subtenon triamcinolone acetonide injection

**Table 2. Analyses of IOP elevation (IOP > 21 mmHg).**

| | Total | IOP elevation | No IOP elevation | P-value |
|---|---|---|---|---|
| Number of patients | 1406 eyes | 206 eyes (14.7%) | 1200 eyes (85.3%) | |
| Age | 63.8 ± 12.9 years | 57.6 ± 12.6 years | 64.9 ± 12.7 years | *<**0.0001*** |
| Sex (male) | 686 (54.8%) | 116 (56.3%) | 642 (53.5%) | 0.45 |
| Baseline IOP | 14.2 ± 3.1 mmHg | 16.4 ± 2.5 mmHg | 13.8 ± 3.0 mmHg | *<**0.0001*** |
| Highest IOP | 18.1 ± 5.0 mmHg | 26.6 ± 6.2 mmHg | 16.6 ± 2.9 mmHg | *<**0.0001*** |
| Total steroid dose | 20.5 ± 4.3 mg | 23.3 ± 10.9 mg | 20.0 ± 0.0 mg | *<**0.0001*** |
| Causative diseases | | | | |
| DME | 632 eyes (45.0%) | 89 eyes (43.2%) | 543 eyes (45.3%) | 0.63 |
| RVO | 457 eyes (32.5%) | 65 eyes (31.6%) | 392 eyes (39.0%) | |
| Uveitis | 223 eyes (15.9%) | 39 eyes (18.9%) | 184 eyes (15.3%) | |
| Others | 94 eyes (6.7%) | 13 eyes (6.3%) | 81 eyes (6.8%) | |
| Treatments | | | | |
| Eye drops | | 106 eyes (51.5% of eyes with IOP elevation; 7.5% of all eyes) | 0 eyes | *<**0.0001*** |
| Glaucoma surgery | | 2 eyes (1.0% of eyes with IOP elevation; 0.14% of all eyes) | 0 eyes | |

DME, diabetic macular oedema; IOP, intraocular pressure; RVO, retinal vein occlusion

and glaucoma surgery was performed in two eyes (1.0% of eyes with IOP elevation, 0.14% of all). After treatment, IOP returned to normal in patients who had exhibited IOP elevation. Sex and causative diseases were not associated with IOP elevation.

## Secondary endpoint: IOP increase > 6 mmHg

Analyses of IOP increase ≥ 6 mmHg are shown in Table 3. Briefly, the incidence of IOP increase ≥ 6 mmHg was 23.3% (328 eyes). The age of patients with IOP increase ≥ 6 mmHg was significantly younger than that of patients with IOP increase < 6 mmHg (p<0.0001, Wilcoxon signed-rank test). Baseline and highest IOP, total steroid dose, and the incidence of IOP elevation were significantly higher in patients with IOP increase ≥ 6 mmHg than in patients with IOP increase < 6 mmHg (each p<0.0001, Wilcoxon signed-rank test). As causative diseases, the incidences of DME and uveitis were significantly higher in patients with IOP increase ≥ 6 mmHg than in patients with IOP increase < 6 mmHg (p = 0.02, Fisher's exact test). For treatments in patients with IOP increase ≥ 6 mmHg, eye drops were administered in 95 eyes (29.0% of eyes with IOP increase > 6 mmHg, 6.8% of all eyes) and glaucoma surgery was performed in two eyes (1.2% of eyes with IOP increase ≥ 6 mmHg, 0.14% of all eyes). After treatment, IOP returned to normal in patients who had exhibited IOP increase ≥ 6 mmHg. Sex was not associated with IOP increase ≥ 6 mmHg.

## Other endpoints

Regarding causative diseases, the incidences of IOP elevation (IOP > 21 mmHg) and IOP increase ≥ 6 mmHg in patients with DME, RVO, uveitis, and others are shown in Tables 2 and 3, respectively. No causative diseases were associated with IOP elevation (p = 0.64, Fisher's exact test); however, the incidences of DME and uveitis were significantly associated with IOP increase ≥ 6 mmHg (p = 0.02, Fisher's exact test).

To assess risk factors for IOP elevation and IOP increase without the effect of steroid dose, we analysed the data with a fixed steroid dose (TA = 20 mg) (Table 4). Similar to the findings with a variable steroid dose, younger age and higher baseline IOP were associated with IOP

**Table 3. Analyses of IOP increase ≥ 6 mmHg.**

| | Total | IOP increase ≥ 6 mmHg | IOP increase < 6 mmHg | P-value |
|---|---|---|---|---|
| Number of patients | 1406 eyes | 328 eyes (23.3%) | 1078 eyes (76.7%) | |
| Age | 63.8 ± 12.9 years | 60.9 ± 14.0 years | 64.7 ± 12.5 years | <**0.0001** |
| Sex (male) | 686 (54.8%) | 191 (58.2%) | 567 (52.6%) | 0.07 |
| Baseline IOP | 14.2 ± 3.1 mmHg | 13.5 ± 3.1 mmHg | 14.4 ± 3.0 mmHg | <**0.0001** |
| Highest IOP | 18.1 ± 5.0 mmHg | 23.0 ± 6.7 mmHg | 16.6 ± 3.1 mmHg | <**0.0001** |
| Total steroid dose | 20.5 ± 4.3 mg | 21.9 ± 8.6 mg | 20.1 ± 1.1 mg | <**0.0001** |
| Causative diseases | | | | |
| DME | 632 eyes (45.0%) | 135 eyes (41.2%) | 497 eyes (32.8%) | **0.02** |
| RVO | 457 eyes (32.5%) | 103 eyes (31.4%) | 354 eyes (39.0%) | |
| Uveitis | 223 eyes (15.9%) | 70 eyes (21.3%) | 153 eyes (14.2%) | |
| Others | 94 eyes (6.7%) | 20 eyes (6.1%) | 74 eyes (6.9%) | |
| Incidence of IOP elevation | 206 eyes | 161 eyes (78.2%) | 45 eyes (21.8%) | <**0.0001** |
| Treatments | | | | |
| Eye drops | 106 eyes | 95 eyes (29.0% of eyes with IOP increase ≥ 6 mmHg; 6.8% of all eyes) | 11 eyes (1.0%) | <**0.0001** |
| Glaucoma surgery | 2 eyes | 2 eyes (1.2% of eyes with IOP increase ≥ 6 mmHg; 0.14% of all eyes) | 0 eyes (0.0%) | |

DME, diabetic macular oedema; IOP, intraocular pressure; RVO, retinal vein occlusion

elevation. In addition, IOP increase ≥ 6 mmHg was associated with younger age, lower baseline IOP, higher incidence of uveitis, and lower incidence of DME.

## Discussion

In this study, we analysed the proportions of eyes with IOP elevation after STTA in a sample of more than 1,000 Japanese patients, and assessed the characteristics associated with IOP

**Table 4. Analyses of IOP elevation and IOP increase with a fixed steroid dose (TA = 20 mg).**

| | Total | IOP elevation | No IOP elevation | P-value |
|---|---|---|---|---|
| Number of patients | 1383 eyes | 183 eyes (13.2%) | 1200 eyes (86.8%) | |
| Age | 63.8 ± 12.9 years | 57.1 ± 12.7 years | 64.9 ± 12.7 years | <**0.0001** |
| Baseline IOP | 14.2 ± 3.1 mmHg | 16.4 ± 2.5 mmHg | 13.8 ± 3.0 mmHg | <**0.0001** |
| Highest IOP | 18.0 ± 4.9 mmHg | 26.7 ± 6.4 mmHg | 16.6 ± 2.9 mmHg | <**0.0001** |
| Causative diseases | | | | |
| DME | 626 eyes (45.3%) | 83 eyes (45.4%) | 543 eyes (45.3%) | 0.38 |
| RVO | 443 eyes (32.0%) | 51 eyes (27.9%) | 392 eyes (32.7%) | |
| Uveitis | 220 eyes (15.9%) | 36 eyes (19.7%) | 184 eyes (15.3%) | |
| Others | 94 eyes (6.8%) | 13 eyes (7.1%) | 81 eyes (6.8%) | |
| | Total | IOP increase ≥ 6 mmHg | IOP increase < 6 mmHg | |
| Number of patients | | 308 eyes (22.3%) | 1075 eyes (77.7%) | |
| Age | | 57.1 ± 12.7 years | 64.9 ± 12.7 years | <**0.0001** |
| Baseline IOP | | 13.4 ± 3.1 mmHg | 14.4 ± 3.0 mmHg | <**0.0001** |
| Highest IOP | | 22.7 ± 6.8 mmHg | 16.6 ± 3.1 mmHg | <**0.0001** |
| Causative diseases | | | | |
| DME | | 129 eyes (41.9%) | 497 eyes (46.2%) | **0.01** |
| RVO | | 91 eyes (29.6%) | 352 eyes (32.7%) | |
| Uveitis | | 68 eyes (22.1%) | 152 eyes (14.1%) | |
| Others | | 20 eyes (6.5%) | 74 eyes (6.9%) | |

DME, diabetic macular oedema; IOP, intraocular pressure; RVO, retinal vein occlusion

elevation in those patients. IOP elevation and IOP increase ≥ 6 mmHg were found 14.7% and 23.3% of patients who received STTA, respectively. These results are similar to the findings of a prior study, in which IOP elevation (IOP > 21 mmHg) was observed in 10/62 eyes (16%) following treatment with dexamethasone eye drops [38]. In contrast, IOP elevation after treatment with 4 mg IVTA was observed in 48/150 eyes (32%) [40] and 26/60 eyes (43%) [39].

As in the present study, IOP elevation after treatment with 20 mg STTA was observed in 7/85 eyes (8.2%) [7] and 8/48 eyes (16.7%) [34]; IOP elevation after treatment with 40 mg STTA was observed in 6/49 eyes (12%) [41], 6/35 eyes (17%) [5], and 26/115 eyes (22.6%) [42].

Prior studies comparing IVTA and STTA revealed no eyes with IOP elevation (IOP > 25 mmHg) and no significant differences in IOP between the two groups for during a 6 month follow-up period [3]. Moreover, IVTA was reported to cause significantly higher IOP elevation, compared with that caused by STTA [4]. According to Inatani and colleagues, IOP elevation > 24 mmHg was observed in 2.8%, 3.7%, and 13.5% of patients after treatment with 12 mg, 20 mg, and 40 mg STTA, respectively; they concluded that the risk of IOP elevation was increased by TA treatment in a dose-dependent manner [35].

The mechanism of steroid-induced IOP elevation remains unclear. In electron microscopy-based anatomical analyses of the trabecular meshwork in patients with steroid glaucoma who underwent trabeculectomy, the trabecular meshwork exhibited morphological changes secondary to deposits of fibrillary elements and extracellular material; thus, the aqueous humour route had become obstructed [43]. Reduced phagocytosis activity in trabecular meshwork cells led to increased aqueous humour flow resistance [44]. Whereas some researchers reported an association between the $^{MYOC}$ gene and steroid-induced glaucoma [45, 46], others reported no such association [47, 48]. Finally, the $^{MYOC}$ genetic mutation was negatively associated with steroid-induced glaucoma in a recent study [49].

When comparing patients with bilateral STTA and those with unilateral STTA, patients with unilateral STTA were older and exhibited a higher incidence of RVO than patients with bilateral STTA. This is reasonable because RVO is known to develop in elderly people. However, baseline and highest IOP, as well as the incidences of IOP elevation and IOP increase ≥ 6 mmHg, were not associated with bilateral or unilateral STTA. Regarding risk factors for IOP elevation, younger people are widely known to be at higher risk [34–37]. Moreover, DME and uveitis contributed to IOP elevation in the present study, as previously reported. Yamamoto and colleagues reported that the average age of patients who received STTA and exhibited IOP < 24 mmHg was 65.8 years, while the average age of patients who received STTA and exhibited IOP > 24 mmHg was 57.1 years. Baseline IOP values in these patients were 13.8 mmHg and 15.1 mmHg, respectively, suggesting that younger age and higher baseline IOP were risk factors for IOP elevation [37]. The results from our study with a large number of patients were similar; younger age and higher baseline IOP were risk factors for IOP elevation. Regarding high baseline IOP, it is reasonable that IOP is elevated after STTA in patients with higher baseline IOP; however, it remains unclear what is an appropriate IOP difference between baseline and highest IOP after STTA. Patients with IOP increase ≥ 6 mmHg after STTA are classified as steroid responders [34, 50–52], suggesting that patients with low baseline IOP are more frequently steroid responders. In addition, the risk factors for IOP increase ≥ 6 mmHg in our study were similar to those of prior studies: younger age, lower baseline IOP, steroid dose, and higher incidences of DME and uveitis.

To assess risk factors for IOP elevation and IOP increase without the effect of steroid dose, we analysed the data with a fixed steroid dose (TA = 20 mg). From the analyses of IOP elevation under the steroid fixed condition, the risk factors were same; younger age and higher baseline IOP. Similar to the findings with a variable steroid dose, the risk factors for IOP increase ≥ 6 mmHg were younger age, lower baseline IOP, and higher incidence of uveitis.

However, a lower incidence of DME was identified as a risk factor with a fixed steroid dose, whereas a higher incidence of DME was a risk factor under a variable steroid dose; this suggested that IOP might increase in association with the dose of TA, especially in patients with DME.

An absolute IOP elevation (i.e., IOP > 21 mmHg), rather than a relative difference in IOP (i.e., IOP increase ≥ 6 mmHg), is evaluated in clinical treatment. However, a relatively large difference in IOP may induce an effect on ganglion cells. In the present study, 14.7% and 23.3% of all patients showed IOP elevation (IOP > 21 mmHg) and IOP increase ≥ 6 mmHg after STTA, respectively. In patients with IOP elevation, steroid dose was associated with IOP elevation. This steroid dose-dependent elevation in IOP was previously reported [35]. Therefore, patients who undergo multiple STTA treatments need strict IOP observation. Furthermore, patients who experienced subtenon overflow of TA developed IOP elevation at a high rate after STTA [9]. Therefore, to prevent IOP elevation, it is necessary to perform STTA in a precise manner.

We acknowledge several limitations to this study. First, the IOP measurement device was not standardized among hospitals. However, IOP measurements in each patient were performed with an identical device during the follow-up period, suggesting that the analysis of IOP increase ≥ 6 mmHg might have yielded more representative data than that produced by analysis of IOP elevation. Second, the time to measure IOP was not standardized among hospitals because our study was a retrospective multicentre study. However, this study included a large number of patients, such that the respective risks of IOP elevation and increase are likely to be accurate and representative.

In summary, we retrospectively examined the incidences of IOP elevation and IOP increase ≥ 6 mmHg in a large number of Japanese patients who received STTA. IOP elevation and IOP increase ≥ 6 mmHg were found in 14.7% and 23.3% of patients who received STTA, respectively. The risk factors of IOP elevation were younger age, higher baseline IOP, and steroid dose; risk factors of IOP increase ≥ 6 mmHg were younger age, lower baseline IOP, steroid dose, and higher incidences of DME and uveitis. Thus, in patients receiving STTA, IOP should be closely monitored in those with younger age, higher baseline IOP, and higher incidences of DME and uveitis.

## Supporting information

**S1 Dataset. Dataset.**
(PDF)

## Acknowledgments

We thank Tadanobu Yoshikawa, M.D., Ph.D. for data acquisition, and Ryan Chastain-Gross, Ph.D., from Edanz Group (www.edanzediting.com/ac) for editing a draft of this manuscript.

## Author Contributions

**Conceptualization:** Hiroto Ishikawa, Fumi Gomi.

**Data curation:** Yuki Maeda, Miho Shimizu, Rie Ogihara, Chihiro Yamanaka, Masahiko Sugimoto, Yoshihiro Takamura, Nahoko Ogata, Fumi Gomi.

**Formal analysis:** Yuki Maeda, Hiroto Ishikawa.

**Investigation:** Hiroto Ishikawa, Fumi Gomi.

**Methodology:** Hiroto Ishikawa.

**Project administration:** Hiroto Ishikawa.

**Resources:** Yuki Maeda.

**Supervision:** Hiroki Nishikawa, Miho Shimizu, Takamasa Kinoshita, Rie Ogihara, Shigehiko Kitano, Chihiro Yamanaka, Yoshinori Mitamura, Masahiko Sugimoto, Mineo Kondo, Yoshihiro Takamura, Nahoko Ogata, Tomohiro Ikeda, Fumi Gomi.

**Validation:** Hiroki Nishikawa, Takamasa Kinoshita, Shigehiko Kitano, Yoshinori Mitamura, Mineo Kondo, Yoshihiro Takamura, Nahoko Ogata, Tomohiro Ikeda, Fumi Gomi.

**Writing – original draft:** Yuki Maeda, Hiroto Ishikawa, Hiroki Nishikawa, Fumi Gomi.

**Writing – review & editing:** Hiroto Ishikawa, Hiroki Nishikawa, Takamasa Kinoshita, Shigehiko Kitano, Yoshinori Mitamura, Mineo Kondo, Yoshihiro Takamura, Nahoko Ogata, Tomohiro Ikeda, Fumi Gomi.

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
