## [Decision Letter · Decision Letter 0]

31 Oct 2019

PONE-D-19-24856

Intraocular pressure elevation after subtenon triamcinolone acetonide injection; Multicentre retrospective cohort study in Japan

PLOS ONE

Dear Dr Ishikawa,

Thank you for submitting your manuscript to PLOS ONE. After careful consideration, we feel that it has merit but does not fully meet PLOS ONE’s publication criteria as it currently stands. Therefore, we invite you to submit a revised version of the manuscript that addresses the points raised during the review process.

The expert reviewer indicated that your results do not fully support your conclusions and that you have not made all of your data available. Pleas indicate how many patients had received subtenon triamcinolone acetonide injection prior to exclusion, for example.

We would appreciate receiving your revised manuscript by Dec 15 2019 11:59PM. To enhance the reproducibility of your results, we recommend that if applicable you deposit your laboratory protocols in protocols.io, where a protocol can be assigned its own identifier (DOI) such that it can be cited independently in the future. For instructions see: http://journals.plos.org/plosone/s/submission-guidelines#loc-laboratory-protocols

We look forward to receiving your revised manuscript.

Kind regards,

Alfred S Lewin, Ph.D.

Academic Editor

PLOS ONE

Journal Requirements:

Reviewers' comments:

Reviewer's Responses to Questions

**Comments to the Author**

1. Is the manuscript technically sound, and do the data support the conclusions?

Reviewer #1: Partly

2. Has the statistical analysis been performed appropriately and rigorously? 

Reviewer #1: Yes

3. Have the authors made all data underlying the findings in their manuscript fully available?

Reviewer #1: No

4. Is the manuscript presented in an intelligible fashion and written in standard English?

Reviewer #1: Yes

5. Review Comments to the Author

Reviewer #1: This paper is important since it addresses the real possibility of iatrogenic disease. However, many issues remain,

Abstract.

a. Results: Higher incidence of DME and uveitis are listed as risk factors for raised IOP. Yet in the next line it is mentioned that low incidence of DME is associated with raised IOP. Please clarify

2. Methods

a. Would be interesting to know how many patients had received STTA in the study period prior to exclusion due to the criteria given.

b. What were the specific diagnosis of all subjects reeving SSTA?

c. Exclusion criteria were exclusion from SSTA or exclusion from the study. Please clarified.

3. Results

a. It would be interesting to know what was the IOP elevation in each disease group

b. In 23% of cases, the IOP rose > 6 mmHg. How were they all treated? What was the disc like?

c. In how many did the IOP return back to normal? Did the bolus require to be removed?

d. Could not find a correlation table in your manuscript to say the IOP is correlated to the underlying disease. A regression analysis would be an appropriate tool.

6. PLOS authors have the option to publish the peer review history of their article (what does this mean?). If published, this will include your full peer review and any attached files.

Reviewer #1: No

---

## [Author Response · Author response to Decision Letter 0]

11 Nov 2019

11 November 2019

Dr. Alfred S. Lewin

Academic Editor

PLOS ONE

Dear Dr. Lewin,

My coauthors and I wish to thank you for allowing us to re-submit our manuscript, entitled ‘Intraocular pressure elevation after subtenon triamcinolone acetonide injection; Multicentre retrospective cohort study in Japan’ (PONE-D-19-24856).

We appreciate the reviewer’s valuable comments regarding our manuscript. We have revised our manuscript and addressed the concerns expressed by the reviewer in a point-by-point manner. The revised or newly added text is highlighted in yellow. Our responses to each of the reviewer’s comments are provided below.

Referee(s)' Comments to Author:

Reviewer #1: This paper is important since it addresses the real possibility of iatrogenic disease. However, many issues remain,

1. Abstract.

a. Results: Higher incidence of DME and uveitis are listed as risk factors for raised IOP. Yet in the next line it is mentioned that low incidence of DME is associated with raised IOP. Please clarify

Response:

Younger age, lower baseline IOP, steroid dose, and higher incidence of DME were associated with raised IOP. However, when we fixed the steroid dose at 20 mg, we found that a lower incidence of DME was associated with raised IOP (only when the steroid dose was fixed as mentioned here), suggesting that IOP might increase in association with the steroid dose, especially in patients with DME. Please note that we mentioned this in the Discussion section in the original manuscript.

2. Methods

a. Would be interesting to know how many patients had received STTA in the study period prior to exclusion due to the criteria given.

Response:

We thank the reviewer for this important comment. Unfortunately, we could not determine the total number of enrolled patients prior to exclusion, because each institution only provided data from patients after exclusion had been performed.

b. What were the specific diagnosis of all subjects reeving SSTA?

Response:

We mentioned causative diseases in the original manuscript. In the revised version, we have added the following additional details: ‘Causative diseases were DME (632 eyes, 45.0%), RVO (457 eyes, 32.5%), uveitis (223 eyes, 15.9%), and others (94 eyes, 6.7%), including optic neuritis, Thyroid-associated ophthalmopathy, and Irvine-Gass Syndrome’.

c. Exclusion criteria were exclusion from SSTA or exclusion from the study. Please clarified.

Response:

These were exclusion criteria for the present study. We have added the following text (highlighted in yellow) in the Material and Methods section: ‘Exclusion criteria for this study were: 1) existing glaucoma diagnosis and associated medication; 2) baseline IOP > 21 mmHg; 3) any intraocular surgery except glaucoma surgery within 6 months after STTA injection. A total of 1406 eyes were analysed in 1252 Japanese patients who received STTA.’

3. Results

a. It would be interesting to know what was the IOP elevation in each disease group

Response:

In accordance with the reviewer’s comment, we have analysed the IOP values in each disease group and added these in the Results section as follows: ‘The highest IOP values in patients with DME, RVO, uveitis, and others were 18.2 ± 4.5 mmHg, 18.1 ± 5.4 mmHg, 18.1 ± 5.7 mmHg, and 17.3 ± 4.8 mmHg, respectively.’

b. In 23% of cases, the IOP rose > 6 mmHg. How were they all treated? What was the disc like?

Response:

We included this information in the Results section and Table 3 in the original manuscript, as follows: ‘For treatments in patients with IOP increase ≥ 6 mmHg, eye drops were administered in 95 eyes (29.0% of eyes with IOP increase > 6 mmHg, 6.8% of all eyes) and glaucoma surgery was performed in two eyes (1.2% of eyes with IOP increase ≥ 6 mmHg, 0.14% of all eyes).’

c. In how many did the IOP return back to normal? Did the bolus require to be removed?

Response:

All instances of IOP elevation were resolved after treatment. We have added the following explanation in the Results section: ‘After treatment, IOP returned to normal in patients who had exhibited IOP elevation.’ and ‘After treatment, IOP returned to normal in patients who had exhibited IOP increase ≥ 6 mmHg.’.

d. Could not find a correlation table in your manuscript to say the IOP is correlated to the underlying disease. A regression analysis would be an appropriate tool.

Response:

We thank the reviewer for this important suggestion. IOP is a continuous variable, whereas causative diseases were represented by a discrete variable. We used Fisher’s exact test for analyses of the association between IOP and causative diseases. Therefore, we calculated p-values as shown in Tables 1 – 4.

Please address all correspondence to:

Hiroto Ishikawa

Department of Ophthalmology, Hyogo College of Medicine

1-1, Mukogawa-cho, Nishinomiya, Hyogo 663-8501, Japan 

Tel; +81-798-45-6462, Fax; +81-798-45-6464, e-mail; ohmyeye@gmail.com

Thank you for your time in reviewing our re-submission. We look forward to hearing from you at your earliest convenience.

Yours sincerely,

Hiroto Ishikawa

---

## [Editor Report · Decision Letter 1]

20 Nov 2019

Intraocular pressure elevation after subtenon triamcinolone acetonide injection; Multicentre retrospective cohort study in Japan

PONE-D-19-24856R1

Dear Dr. Ishikawa,

We are pleased to inform you that your manuscript has been judged scientifically suitable for publication and will be formally accepted for publication once it complies with all outstanding technical requirements.

With kind regards,

Alfred S Lewin, Ph.D.

Section Editor

PLOS ONE
---

## [Editor Report · Acceptance letter]

26 Nov 2019

PONE-D-19-24856R1 

Intraocular pressure elevation after subtenon triamcinolone acetonide injection; Multicentre retrospective cohort study in Japan 

Dear Dr. Ishikawa:

I am pleased to inform you that your manuscript has been deemed suitable for publication in PLOS ONE. Congratulations! Your manuscript is now with our production department. 

With kind regards,

on behalf of

Dr. Alfred S Lewin 

Section Editor

PLOS ONE